# The Role of miRNAs in Childhood Acute Lymphoblastic Leukemia Relapse and the Associated Molecular Mechanisms

**DOI:** 10.3390/ijms25010119

**Published:** 2023-12-21

**Authors:** Dalia Barrios-Palacios, Jorge Organista-Nava, Juan Carlos Balandrán, Luz del Carmen Alarcón-Romero, Ma Isabel Zubillaga-Guerrero, Berenice Illades-Aguiar, Alinne Ayulieth Rivas-Alarcón, Jessica Julieth Diaz-Lucas, Yazmín Gómez-Gómez, Marco Antonio Leyva-Vázquez

**Affiliations:** 1Laboratorio de Biomedicina Molecular, Facultad de Ciencias Químico Biológicas, Universidad Autónoma de Guerrero, Chilpancingo 39090, Guerrero, Mexico; daliabarriospalacios@gmail.com (D.B.-P.); jorgeorganista@uagro.mx (J.O.-N.); billades@uagro.mx (B.I.-A.); alinnerivas@uagro.mx (A.A.R.-A.); 19434847@uagro.mx (J.J.D.-L.); 2Department of Pathology and Laura and Isaac Perlmutter Cancer Center, NYU School of Medicine, New York, NY 10016, USA; jcbalandran@gmail.com; 3Laboratorio de Citopatología, Facultad de Ciencias Químico Biológicas, Universidad Autónoma de Guerrero, Chilpancingo 39090, Guerrero, Mexico; lcalarcon@uagro.mx (L.d.C.A.-R.); mzubillaga@uagro.mx (M.I.Z.-G.)

**Keywords:** ALL, miRNAs, relapse, resistance to therapy

## Abstract

Acute lymphoblastic leukemia (ALL) is the most common cancer in children worldwide. Although ALL patients’ overall survival rates in wealthy countries currently surpass 80%, 15–20% of patients still experience relapse. The underlying mechanisms of relapse are still not fully understood, and little progress has been made in treating refractory or relapsed disease. Disease relapse and treatment failure are common causes of leukemia-related death. In ALL relapse, several gene signatures have been identified, but it is also important to study miRNAs involved in ALL relapse in an effort to avoid relapse and to achieve better survival rates since miRNAs regulate target genes that participate in signaling pathways involved in relapse, such as those related to drug resistance, survival signals, and antiapoptotic mechanisms. Several miRNAs, such as miR-24, miR-27a, miR-99/100, miR-124, miR-1225b, miR-128b, miR-142-3p, miR-155 and miR-335-3p, are valuable biomarkers for prognosis and treatment response in ALL patients. Thus, this review aimed to analyze the primary miRNAs involved in pediatric ALL relapse and explore the underlying molecular mechanisms in an effort to identify miRNAs that may be potential candidates for anti-ALL therapy soon.

## 1. Introduction

The most common type of pediatric cancer, acute lymphoblastic leukemia (ALL), accounts for approximately 25% of all pediatric cancers globally [1]. Eighty percent of cases involve children, and 20% involve adults; thus, this disease is known primarily as pediatric leukemia [2]. Optimizing risk-directed therapeutic strategies and intensive protocols for ALL treatment has achieved remarkable progress in the last six decades. Presently, event-free and overall survival (OS) rates for ALL patients in developed nations are greater than 80% and 90%, respectively [1,3]. However, relapse still occurs in 15–20% of children, and patients who experience recurrence have an OS rate of less than 10% [4,5]. 

Regardless of better survival, the costs of chemotherapy for ALL and illness-related expenses are immense for these patients and families [6] with adverse economic consequences for families in developed countries but even more severe effects in developing countries. Additionally, direct costs include expenditures related to illness, such as those associated with transport, food and accommodation; indirect costs include the value of productivity loss, such as cutting on work time, taking unpaid leave or quitting one’s job. In Canada, average treatment costs for childhood ALL were estimated to be between CAD 239,000 (adjusted to 2018 Canadian dollars) [7]), while in the United States, they were estimated to be US$ 483,000 (adjusted to 2020 US dollars) [8]; in addition, in China, they were estimated to be between US$ 15,128 and US$ 45,386 in 2018. The high cost of treatment can cause patients give up treatment [9].

Little progress has been made in treating people with relapsed leukemia, and the mechanism of leukemia relapse is still not entirely understood [10,11]. Patients who are nonresponders or patients with relapsed disease have a dire prognosis, as relapse and treatment failure are among the most common causes of leukemia-related death [12,13]. Therefore, it is critical to pinpoint the underlying mechanisms to prolong the survival of patients who experience relapse.

MicroRNAs (miRNAs) are single-stranded noncoding RNAs of approximately 18–25 nucleotides that suppress protein translation from mRNA or encourage its destruction to adversely regulate gene expression at the posttranscriptional phase [13,14]. The cell needs these molecules to operate normally. They control several biological processes, including hematopoiesis, the cell cycle, apoptosis, and drug sensitivity [3,15].

In 2002, Calin et al. reported that miRNAs could be essential for cancer development when they discovered that miR-15 and miR-16 were downregulated or deleted in almost 70% of chronic lymphocytic leukemia patients [16]. The formation and progression of other human cancers have also been linked to miRNA expression, and specific miRNA signatures have been linked to patient outcomes or therapeutic response [11]. Many studies have also reported the association between miRNA expression and patient prognosis [1,15,17,18,19]. Han et al. reported several miRNAs that were associated with glucocorticoid response and risk of relapse [17], and Nemes et al. reported a correlation between overexpression of miR-128b and a poor prognosis, poor response to prednisolone and worse survival in pediatric patients [18]. Additionally, Liang et al. reported that miR-155 was associated with cellular proliferation and a poor prognosis [19]. Furthermore, several miRNA signatures have been identified as better prognostic factors than conventional signatures in leukemia [20]. However, the functional role of miRNAs and the molecular mechanism involved in ALL relapse have not been fully elucidated. Thus, in this review, we identified the primary miRNAs linked to ALL relapse function and molecular mechanisms.

## 2. Acute Lymphoblastic Leukemia

ALL is a cancerous condition that develops from hematopoietic progenitors composed of T cells (20–25%) or B cells (80–85%). Genetic mutations that prevent lymphoid development can be acquired by lymphoid precursor cells, resulting in aberrant cell death and survival [5,21]. The presence of aberrant, immature lymphoid precursor cells in the bone marrow (BM) and the clonal growth of these cells are hallmarks of ALL. These aberrant cells can also spread through systemic circulation from the BM to the entire body, where they replace healthy hematopoietic cells [4]. ALL is a heterogeneous disease that comprises different immunophenotypic subtypes and is subclassified by chromosomal and molecular abnormalities and gene expression patterns [22]. Among the ALL immunophenotype subtypes, three large groups can be distinguished: (1) pre-B-cell ALL, (2) mature B-cell ALL and (3) T-cell ALL. Pre-B-cell ALL is characterized by the expression of immunoglobulins. Cytoplasmic (cIg) markers include CD79a, CD19, HLA-DR and CD10. Mature B-cell ALL is characterized by the expression of surface immunoglobulins (sIg) and μ heavy chains. Finally, T-cell ALL is characterized by the expression of cytoplasmic CD3, CD7, CD5, or CD2 [23]. Among the chromosomal and molecular abnormalities observed in ALL patients, the clinically relevant subgroups were BCR-ABL1, ETV6-RUNX1, and E2A-PBX1 [24]. Finally, regarding the gene expression patterns in ALL, Philadelphia chromosome-positive (Ph+) ALL was correlated with the presence of ABL1 mutations and IKZF1 deletion. Among the BCR/ABL-like genes, the correlated genes were IGH@CRLF2, NUP214-ABL1, EBF1-PDGFRB, BCRJAK2, STRN3-JAK2, IGH@-EPOR, ∆-CRLF2 and IKZF1 deletion [25]. In ALL with TCR translocations and various oncogenes t(1;14) t(10;14) t(5;14), the genes involved are LMO1, LMO2, TAL1, TLX1, and TLX3, while in the Del(1)(p32) translocation, SIL-TAL1 is the gene involved. The MLL gene is involved in 11q23 rearrangement. In the t(9;9)(q34;q34) translocations, NUP214-ABLs are involved, while in the t(9;14)(q34;q32) translocations, the genes involved are EML1-ABLs. Additionally, in the ALL early-T precursor, the ETV6, IDH1, IDH2, DNMT3A, FLT3, NRAS, JAK3 and IKZF1 genes are involved [26].

ALL incidence exhibited a bimodal distribution with the first peak occurring in childhood and the second peak occurring around the age of 50. While 80% of ALL cases occur in children, 20% occur in adults [2,27].

## 3. Pediatric Acute Lymphoblastic Leukemia Therapy

The three stages of treatment—induction, consolidation, and maintenance—involve the application of central nervous system (CNS)-directed therapy. Vincristine, anthracycline, corticosteroids, and asparaginase are all part of induction therapy. Cytarabine, methotrexate, anthracyclines, and alkylating drugs were used during consolidation. Patients frequently receive maintenance chemotherapy, which includes 6-mercaptopurine, methotrexate with occasional vincristine, corticosteroids, and intrathecal therapy, after finishing the consolidation phase of treatment [5]. Childhood ALL treatment relies on a risk-based approach (Table 1) [28,29].

## 4. Therapeutic Response

One month after anti-ALL therapy is initiated, patients may experience complete remission or relapse. Absolute neutrophil counts less than 1 × 10^9^/L, platelet counts greater than 100 × 10^9^/L, and MRD levels less than 0.01% are needed for complete remission. Blasts must also be present in less than 5% of the BM aspirate, be absent from peripheral blood, and not invade extramedullary tissues [30]. On the other hand, relapse is indicated by the development of extramedullary leukemia, the presence of 5% BM lymphoblasts, the return of circulating leukemic blasts in the peripheral blood, or an MRD of 0.01% [31]. Three prognostic criteria are used to categorize and stratify relapses: the time to relapse, the location of the relapse, and the immunophenotype (B or T). The timing of relapse is the most crucial prognostic factor. The prognosis is poorer when relapse occurs earlier. The anatomical site of relapse is the second crucial determinant after timing. Patients with intramedullary resection alone or mixed regression had better and intermediate prognoses, whereas patients with BM relapse had the poorest prognosis. Finally, the T-cell immunophenotype was once linked to a poor prognosis, but the emergence of better classification systems has decreased its prognostic value [4].

## 5. Mechanisms Underlying ALL Relapse

A variety of biological variables influence ALL relapse. Several studies suggest that after chemotherapy, residual leukemic cells that survive could cause relapse [11] due to the emergence of pre-existing drug-resistant subclones found at initial diagnosis or due to the acquisition of genomic lesions [12]. Additionally, other studies have suggested the presence of a common ancestral preleukemic clone that causes relapse [32,33].

Resistance to chemical drugs, which can cause relapse, is one of the main difficulties in treating cancer patients [34], and the loss or mutation of some genes has been identified in ALL relapse. The affected genes include the ATP-binding cassette (ABC) transporter (ABCB1, ABCC1, and ABCG2) gene, glutathione (GSH), nuclear receptor subfamily 3 (NR3C1, NR3C2), tumor protein p53 (TP53), 5′-nucleotidase, cytosolic II (NT5C2), folylpolyglutamate synthase (FPGS), CREB-binding protein (CREBBP), MutS homolog 6 (MSH6), PMS1 homolog 2 (PMS2), nuclear receptor binding SET domain protein 2 (WHSC1), and phosphoribosyl pyrophosphate synthetase (PRPS1, PRPS2) [5,12,35]. These genes can be associated with the efflux of many substrates, such as chemotherapeutic drugs, from the cytoplasm of the cell (the ABC transporters) [34]; alterations in the enzymatic antioxidant defenses in lymphocytes (the GSH gene) [36]; alterations in the glucocorticoid response pathway (the NR3C1 and NR3C2 genes) [4]; cell replication (the TP53 gene) [37]; or nucleotide metabolism (the NT5C2 or the genes PRSP1 and PRPS2, for instance) [38,39].

## 6. Strategies for Relapse Therapy

Until recently, treatment options for relapse therapy were limited to intensive cytotoxic chemotherapy with or without site-directed radiotherapy and allogeneic hematopoietic stem cell transplantation (HSCT). Currently, the current strategies for preventing relapse in ALL include immunotherapies, such as blinatumomab, a bispecific T-cell engager (BiTE), with one arm targeting CD19 on B-ALL blasts and the other arm binding to CD3ζ on T cells. Upon binding, T cells become activated and exert perforin-dependent cytotoxicity against target cells expressing CD19 [40]. Another novel immunotherapy treatment is the use of ozogamicin, a CD22 monoclonal antibody bound to calicheamicin, which is a cytotoxic agent that induces double-strand DNA breaks, cell cycle arrest, and apoptotic cell death [41]. Additionally, a third innovative cellular immunotherapeutic approach is chimeric antigen receptor (CAR) T-cell therapy, which involves the genetic modification of T cells to express CAR-targeting tumor antigens [42]. In addition to these therapies, in recent years, there have been targeted therapies, such as the use of an mTOR inhibitor (everolimus) combined with 4-drug reinduction [43], a proteasome inhibitor (bortezomib) plus 4-drug reinduction [44], a nucleoside analog (nelarabine) plus cyclophosphamide and etoposide [45], and the use of liposomal vincristine [46]; all of these include complete clinical trials.

## 7. Side Effects of Pediatric Acute Lymphoblastic Leukemia Therapy

In pediatric acute lymphoblastic leukemia therapy, several serious side effects, including neurotoxicity, cardiac toxicity, growth impairment, delayed puberty, bone toxicity, hepatic dysfunction, visual changes, obesity, an impact on fertility, and neurocognitive effects, are observed [47,48]. With respect to neurological outcomes, central nervous system-directed therapies, including cranial radiotherapy and/or intrathecal chemotherapy with methotrexate (MTX) and the IT cytosine arabinoside, are established risk factors for impaired cognitive function, especially for younger patients [49]. Additionally, vincristine has been associated with dose-dependent peripheral neuropathy [50]. On the other hand, ALL patients face the risk of developing cardiotoxic effects during treatment, especially after anthracycline therapy, which can lead to congestive heart failure, abnormalities of the heart valve, heart attacks, and heart epithelium inflammation [51]. Additionally, ALL chemotherapy can lead to hormonal deficiency of growth hormone, which is the most commonly observed endocrinopathy following radiation therapy in survivors of ALL due to direct injury to the hypothalamus [47]. In addition, bone toxicity during therapy may result in osteoporosis and fracture [52] due to treatment with high-dose MTX, mercaptopurine, and glucocorticoids [53]. Another adverse effect of chemotherapy is obesity, which has been observed in ALL patients compared with the body mass index of the general population. Additionally, these long-term survivors of ALL could have worse depressive symptoms, pain, and anxiety, and they also engaged in physical activity less frequently, all of which can contribute to obesity. On the other hand, infertility is another adverse effect in male and female survivors of childhood ALL, and treatment with alkylating agents is involved [54]. All these consequences may cause employment and insurance problems and impaired quality of life [55].

## 8. miRNAs Associated with Relapse in ALL

miRNAs affect ALL relapse by controlling numerous genes, and miRNAs participate in various signaling pathways in addition to gene loss or mutation. Several of these most studied miRNAs are associated with ALL relapse, and their molecular mechanisms are described below.

### 8.1. miR-24

Pediatric ALL patients with miR-24 overexpression have lower remission and survival rates than those with downregulated miR-24 [15]. According to previous reports, miR-24 is expressed in CD34+ HSPCs and targets human activin receptor type 1 (ALK4) to regulate proper erythropoiesis [56]. Other molecular targets of miR-24 include proapoptotic proteins (fas-associated factor 1 (FAF-1), Bcl-2-like protein 11 (BIM), apoptotic peptidase activating factor 1 (APAF-1) and caspase 9) and cell cycle proteins; thus, miR-24 regulates apoptosis and cell proliferation and promotes the survival of hematopoietic cells [15]. In the intrinsic apoptosis pathway, procaspase-9 is the initiator of caspases. The consequences of failing to activate caspase-9 are substantial and can result in diseases such as cancer [57]. Additionally, miR-24 controls P27Kip1 and p16INK4a, which are essential regulators of cell cycle progression from the G1 to S phase, as well as APAF-1, which is a molecule implicated in the start of the intrinsic route of apoptosis [58]. Cdk4/cyclin D1 complex activity is inhibited by P16, which leads to Rb dephosphorylation, whereas P27 inhibits the cyclin E/cdk2 complex [59]. The BCL2 family member protein BIM (BCL2L11) is another molecule that miR-24 targets. BIM controls normal tissue homeostasis by interacting with proapoptotic proteins, such as p53 upregulated modulator of apoptosis (PUMA), BCL2-associated agonist of cell death (BAD), and apoptosis regulator BAX (BAX), as well as antiapoptotic proteins, such as BCL2 and BCLXL, and it induces myeloid leukemia cell differentiation protein (MCL1). Moreover, aberrant BIM expression alters normal development and may contribute to cancer [60]. Additionally, miR-24 affects the apoptosis pathway by interacting with the open reading frame region (ORF), affecting Fas-associated factor 1 (FAF1), a factor involved in the death-inducing signaling complex (DISC), and miR-24 interacts with the Fas-associated death domain (FADD) and caspase-8, which are crucial for extrinsic apoptosis pathway activation [61].

These studies confirmed that the overexpression of miR-24 protects leukemic cells from apoptosis, which is a necessary process for maintaining tissue homeostasis. Additionally, chemotherapy and radiotherapy failure are caused by resistance to apoptosis, and the overexpression of antiapoptotic genes increases the risk of carcinogenesis [62]. The overexpression of miR-24 has been found to protect cancer cells from apoptosis in liver, stomach, prostate, and cervical tissues. However, miR-24 induces apoptosis in MCF-7 breast cancer cells by targeting the pro-survival protein BCL-2. These findings demonstrate that the cellular environment affects miR-24 function [63] (Figure 1).

### 8.2. miR-27a

miR-27 is another miRNA linked to ALL relapse, and the downregulation of miR-27a was associated with recurrence in pediatric pre-B-ALL patients. In contrast, patients in the high miR-27a expression group had a greater rate of complete remission and longer relapse-free survival than patients in the low-expression group did [17]. Scheibner et al. (2012) reported that miR-27a was downregulated in acute leukemia cell lines and primary tumor samples compared to hematopoietic stem progenitor cells (HSPCs). Additionally, constitutive miR-27a expression promoted cellular apoptosis by downregulating the antiapoptotic 14-3-3 protein, negatively regulating the proapoptotic proteins BAX and BAD [56]. This was accomplished by expressing miR-27a at levels comparable to those in primary HSPCs. Additionally, P-glycoprotein and genes that affect chemotherapeutic agent sensitivity, such as PUMA, multidrug resistance 1 (MDR1), forkhead box M1 (FOXM1), and multidrug resistance protein 1 (MRP1), are regulated by miR-27a [64,65]. Also, miR-27a overexpression improved doxorubicin sensitivity in the K562 leukemia cell line [64]. According to these findings, miR-27 is a tumor suppressor in acute leukemia and a prospective therapeutic target [56]. However, miR-27a plays an oncogenic role in breast cancer and, by suppressing ZBTB10, promotes tumor growth and spread [66]. The results also showed that depending on the biological setting, miR-27a may function as an oncogene or tumor suppressor [67] (Figure 1).

### 8.3. miR-99 and miR-100

miR-100 and miR-99a were found to be downregulated in patients with T-ALL who also had MLL rearrangement and BCR-ABL fusion gene expression. A poor prognosis was associated with the expression levels of these constructs. Furthermore, dexamethasone-induced apoptosis was increased, and cell proliferation was inhibited in the CCFR-CEM, CEM/C1, and Jurkat cell lines when miR-100 and miR-99a were expressed ectopically. Functional studies have been carried out to determine the function of these miRNAs in vivo. These studies revealed that the miR-99 family targets the insulin-like growth factor 1 receptor (IGF1R) and the mTOR signaling pathways, which are linked to the pathogenesis and progression of ALL. Additionally, IGF1R/mTOR pathway suppression inhibits cell proliferation and kick-starts apoptosis by adversely regulating the expression of the antiapoptotic gene MCL1. Moreover, by enhancing dexamethasone-induced apoptosis in ALL patients, miR-100 and miR-99a serve as tumor suppressors and are prospective therapeutic targets [68]. FK506-binding protein 51 (FKBP51) is another protein that miR-99a and miR-100 are anticipated to target. FKBP51 has been linked to the maintenance of cancerous cell proliferation, malignancy, and therapeutic resistance [69,70]. It has also been noted that FKBP51 is reportedly coupled to the GR–Hsp90 complex in the ligand-unbound state, which decreases hormone binding affinity [71], hinders GR nuclear translocation, and exerts proliferative and antiapoptotic effects through NF-kB activation [72]. GR-dependent signaling occurs following the translocation of GR to the nucleus. Finally, after exposure to dexamethasone, the overexpression of miR-100 and miR-99a decreased FKBP51 protein expression and caused GR activation in the nucleus [68].

However, according to other research, the overexpression of miR-100 and miR-99 is linked to a poor prognosis and resistance to vincristine and daunorubicin [73], and the coexpression of miR-125b with miR-99a or miR-100 causes a significant degree of vincristine resistance (represented by the cell viability rate) in ETV6-RUNX1-positive Reh cells expressing miR-125b [74]. Although these findings contradict each other in terms of the tumor suppressor or oncogenic nature of the function of miR-99/100, they showed that miR-100 and miR-99a are related to leukemogenesis and ALL prognosis. The findings, however, are not entirely conclusive (Figure 1).

### 8.4. miR-124

The overexpression of miR-124 is linked to a poor prognosis and a high probability of relapse in individuals with ALL-T and ALL-B [17]. Additionally, compared to patients with downregulated miR-124, who reacted to glucocorticoids, B-ALL and T-ALL patients with overexpressed miR-124 showed resistance to glucocorticoids and an insufficient response to treatment. Additionally, the prednisone-resistant cell lines CEM/C1 and Jurkat showed substantially greater miR-124 expression than did the prednisone-sensitive CCRF-CEM cell line. In vitro research has also shown that the glucocorticoid receptor NR3C1 is a molecular target of miR-124, which results in treatment resistance. Furthermore, the overexpression of miR-124 induced proliferation and prevented dexamethasone-mediated apoptosis in the prednisone-sensitive CCRF–CEM cell line. These findings support the notion that miR-124 is needed for the glucocorticoid response [70]. Finally, an ontology analysis revealed that miR-124 is related to genes associated with drug transport, multidrug resistance, and the regulation of cellular biosynthesis processes [17] (Figure 1).

### 8.5. miR-125b

One of the miRNAs most frequently linked to ALL relapse is miR-125b. miR-125b expression is downregulated in patients with pre-B ALL and T ALL. However, the level of miR-125b considerably increased after the Berlin–Frankfurt–Munster protocol was applied, and an elevated day 33/diagnosis miR-125b ratio was associated with a poor prognosis [1]. According to a different study, pre-B ALL patients with miR-125b overexpression had a poor prognosis and were resistant to vincristine and daunorubicin treatment [73]. Additionally, the coexpression of miR-125 with miR-100 and miR-99 increased vincristine resistance in the ETV6-RUNX1 translocation-positive Reh cell line [74]. Additionally, Zhou et al. reported that miR-125b decreased the expression of the GRK2 and PUMA proteins, which prevented cell death, in the K562, THP-1, Jurkat, and REH cell lines, contributing to daunorubicin resistance [75].

Other researchers have discovered that miR-125b can promote leukemia by targeting the tumor suppressor interferon regulatory factor 4 (IRF4), which is needed for mature B and T cells [76]. They also discovered that miR-125b promotes cell proliferation and induces the expression of factors related to pluripotency in progenitor B cells by targeting the A+T-rich interaction domain 3A protein (ARID3a) [77]. Additionally, miR-125b can affect T-ALL by targeting TNFAIP3, which is a protein that has been shown to serve as a tumor suppressor in several human B-cell lymphomas. This molecule can enhance B-cell proliferation, differentiation, and the reprogramming of glucose metabolism [78]. BMF, TP53, TP53INP1, and MAPK14 (p38a) are additional miR-125b targets connected to the apoptosis pathway. This information suggested that miR-125b is an oncogene. Patients with elevated miR-125b levels after chemotherapy had a poorer prognosis, as miR-125b overexpression can inhibit the effects of chemotherapeutic drugs that cause apoptosis [1] (Figure 1).

### 8.6. miR-128b

According to several studies, individuals with B-ALL and T-ALL who relapsed had higher levels of miR-128b expression than did those who were in remission [17,18,79]. This miRNA controls the transformation of stem progenitor and multipotent progenitor cells (MPPs) into more mature cells, which are essential for the early stages of hematopoiesis [79,80].

Other studies have demonstrated that the tumor suppressor gene PTEN, whose dysregulation activates the phosphatidylinositol 3-kinase (PI3K)/serine/threonine kinase (AKT) signaling pathway, promoting cell survival and oncogenesis, is regulated by miR-128 [68,81]. Another molecular target of miR-128 is the polycomb ring finger (BMI-1) proto-oncogene, which is a transcription factor needed for leukemia stem cell self-renewal and hematopoietic stem cell (HSC) self-renewal [17,79]. The dysregulation of BMI-1 affects the PI3K–AKT–mTOR signaling pathway, which controls cell division, apoptosis, and senescence. As a result, miR-128 is crucial for the development of tumors in ALL [68]. Although miR-128 has been shown to be downregulated in some malignancies, such as breast cancer, its ectopic expression in the presence of doxorubicin increases DNA damage and death. These findings suggested that miR-128 plays an oncogenic role in ALL but is a tumor suppressor in breast cancer [80] (Figure 1).

### 8.7. miR-142-3p

The overexpression of miR-142-3p was associated with relapse and a noticeably shorter survival duration in T-ALL patients than in those with low miR-142-3p expression. Moreover, miR-142-3p was discovered to be a helpful marker for assessing the prognosis of T-ALL patients [82]. Additionally, miR-142-3p was strongly expressed in the human acute T-leukemic cell lines Jurkat, MOLT-3, MOLT-4, and CCRF/CEM, as well as in the pre-B-ALL cell lines REH, RS4;11, and SEM [82,83]. miR-142 expression was reported to be significantly greater in these cell lines than in healthy subjects’ hematopoietic stem cells or normal T cells.

The mRNA of adenylyl cyclase 9 (AC9), an enzyme that catalyzes the conversion of ATP to cAMP, is another target of miR-142-3p [84]. T-cell proliferation is reduced via the cAMP/protein kinase A (PKA) signaling pathway, which upregulates p27kip1 while downregulating cyclin D3. As a result, miR-142-3p overexpression suppresses the cAMP/PKA pathway and encourages the proliferation of leukemic T cells [82]. Additionally, miR-142-3p targets the mRNA of the glucocorticoid receptor (GR), a receptor that effectively binds glucocorticoids (GCs) and has a DNA binding domain and nucleus-targeted sequence, leading to lymphocyte death mostly through the intrinsic apoptosis pathway [85]. Inhibiting the GC pathway yields T cells that are resistant to intrinsic apoptosis. By targeting the GRa and cAMP/PKA signaling pathways, miR-142-3p promotes leukemogenesis in T-ALL by promoting cell proliferation, resistance to glucocorticoid treatment, and apoptosis evasion [82] (Figure 1).

miR-142-3p is a specific miRNA for hematopoietic cells [83]. This miRNA participates in the growth and differentiation of hematopoietic stem cells and is essential for the maturation of T lymphocytes through interferon regulatory factor 7 (IRF7)-mediated signaling, but its deregulation has been associated with leukemia [18,84].

### 8.8. miR-155

The upregulation of miR-155 has been linked to a poor prognosis in pediatric ALL patients according to research by Liang et al. and Sun et al. [19,86]. By targeting zinc finger protein 238 (ZNF238), a molecule with a potential tumor suppressor function in ALL that is needed for the antiproliferative properties of normal cells, Liang et al. discovered that the overexpression of miR-155 increased cell proliferation in ALL cell lines. However, the methods by which ZNF238 regulates transcriptional activity are still unknown [19]. Additionally, miR-155-5p promoted the growth of ALL cells. By explicitly targeting Casitas B-lineage lymphoma (CBL), which ordinarily degrades the interferon regulatory factor 4 (IRF4) protein through ubiquitination, miR-155 suppresses the production of cyclin-dependent kinase 6 (CDK6), preventing apoptosis [86]. These findings suggest that miR-155 may function as an oncogene in ALL [19] and that miR-155 may be targeted in the future to treat ALL by regulating the CBL-mediated IRF4/CDK6 axis (Figure 1).

### 8.9. miR-335

A low expression level of miR-335 is linked to a worse prognosis, a lower 5-year event-free survival rate, and glucocorticoid resistance in pediatric ALL patients according to a genome-wide miRNA microarray investigation [20]. In contrast to those in the control group, the relapsed patients were found to have lower miR-335-3p levels in a later study. Additionally, drug-resistant patients had lower levels of miR-335-3p than drug-sensitive patients. miR-335 influences the expression of the ABCA3 gene, a member of the ABC transporter family, which can disrupt multidrug resistance pathways [87] and target mitogen-activated protein kinase 1 (MAPK1) to influence pathways involved in prednisone resistance and enhanced apoptosis [20]. These investigations demonstrated the tumor suppressor role of miR-335 in ALL.

According to other research, the miR-335 gene promoter region may undergo excessive methylation and become epigenetically silenced, accounting for the dysregulation of miR-335-3p in cancer [88]. Additionally, the long noncoding RNAs (lncRNAs) nuclear paraspeckle assembly transcript 1 (NEAT1) and metastasis-associated lung adenocarcinoma transcript 1 (MALAT1), which may posttranscriptionally regulate miRNA production, have been found to impact miR-335-3p. NEAT1 and MALAT1 are correlated with poor prognosis and play roles in the development of different cancer types [89,90,91]. In addition, NEAT1 and MALAT1 are overexpressed in pediatric ALL, particularly in patients with multidrug resistance. Furthermore, low levels of miR-335-3p are associated with high expression levels of these lncRNAs [87]. The lncRNA cyclin-dependent kinase inhibitor 2B antisense RNA 1 (CDKN2B-AS1), which promotes carcinogenesis and adriamycin (ADR) resistance in pediatric T-cell ALL, also regulates miR-335-3p. Conversely, CDKN2B-AS1 knockdown inhibited tumor growth. It increased adriamycin (ADR) sensitivity in vivo, at least partially by inhibiting miR-335-3p, illuminating a therapeutic target with great potential for treating young T-ALL patients [92]. According to these findings, the lncRNAs NEAT1, MALAT1, and CDKN2B-AS1 regulate miR-335-3p expression and play a role in the mechanisms underlying ALL relapse (Figure 1).

Several miRNAs have been described as involved in ALL relapse. However, studies have revealed molecular targets for only a few miRNAs described in this manuscript. Table 2 summarizes other miRNAs with aberrant expression in ALL and their molecular targets.

## 9. Therapeutics and miRNAs Involved in ALL Relapse

miRNAs can function as both oncogenes and tumor suppressors. Then, therapy can consist of the use of miRNA mimics to restore the physiological expression of miRNAs that are downregulated, the use of miRNA inhibitors targeted against overexpressed miRNAs in the use of interference-type strategies to silence upregulated miRNAs, and the use of epigenetic drugs like DNA-demethylating agents and histone deacetylase inhibitors, which may also be of potential therapeutic use in the expression of epigenetically silenced miRNAs [96]. miR-24, mir-124, miR-125b, miR-128b, miR-142-3p and miR-155 act as oncomiRs in ALL. An example of the therapies that target the miRNAs was carried out by Agirre et al. (2009). In a group of 353 patients with diagnoses of ALL, researchers observed that the expression of miR-124a was downregulated by hypermethylation of the promoter and histone modifications, thus inducing the upregulation of its target, CDK6, and phosphorylation of retinoblastoma (Rb) and leading to the abnormal proliferation of ALL cells. Cyclin-dependent kinase 6 (CDK6) inhibition by sodium butyrate or PD-0332991 decreased ALL cell growth in vitro, whereas the overexpression of pre-miR124a led to decreased tumorigenicity in an in vivo mouse model [97]. These findings exhibit the possibility of therapeutic strategies for patients with ALL either by the use of drugs that inhibit methylation or histone modifications.

Also, miRNAs has been evaluated in a clinical trial. The drug MRG-106 (cobomarsen) is a miR-155 inhibitor synthesized as a locked nucleic acid (LNA)-modified oligonucleotide. A phase 1 clinical trial (ClinicalTrials.gov identifier: NCT02580552) tested MRG-106 in cutaneous T-cell lymphoma (CTCL), CLL, diffuse large B-cell lymphoma (DLBCL), and adult T-cell leukemia/lymphoma (ATLL) patients. Additionally, a phase 2 clinical trial (ClinicalTrials.gov identifier: NCT03713320) in the CTCL was developed. Unfortunately, it was terminated early due to business reasons of the company. Because the early termination of the phase 2 clinical trial for MRG-106 was not caused by issues of safety or efficacy, this drug may still be tested in further trials and may be promising for clinical application [98]. However, greater efforts are required to translate these laboratory-oriented studies into clinical practice, and this miRNA could provide an important therapeutic strategy for ALL in the future.

## 10. Conclusions and Future Perspectives

Although most newly diagnosed acute lymphoblastic leukemia (ALL) patients have improved outcomes due to therapeutic breakthroughs, refractory or relapsed disease continues to present a substantial clinical challenge [99] since the long-term survival rate of affected patients is low. Additionally, there are still no sensitive prognostic biomarkers for the early detection of refractory/relapsed disease in ALL patients; moreover, these biomarkers allow doctors to monitor patient treatment response, predict patient prognosis [1], and identify those who are likely to experience recurrence [3]. Therefore, new prognostic markers must be found.

Recently, miRNA signatures have been identified as robust prognostic biomarkers in ALL [20]. In this review, we summarize the existing evidence regarding the potential of miRNAs as relapse biomarkers and explore the molecular mechanisms involved in ALL [3,11,100]. miR-24, miR-27a, miR-124, miR-125b, miR-128b, miR-143-3p, miR-155, miR-210, and miR-335 are associated with ALL relapse and drug resistance, cell proliferation, differentiation and apoptosis. However, the above-mentioned mechanisms partially explain why relapse occurs because miRNAs are involved in complex regulatory networks and participate in many biological processes [66]. Additionally, miRNAs can interact with other noncoding RNAs, including lncRNAs and mRNAs involved in various biological processes, such as cell division, the cell cycle, migration, and drug resistance [101,102]. Thus, research on lncRNA–miRNA–mRNA networks is essential for understanding this disease [103] and its association with relapse.

In ALL, genomic imbalances promote the development of the disease [104]. In addition to genomic factors, miRNAs are important epigenetic factors that regulate gene expression through interactions among miRNA–mRNA targets that can promote the subexpression of tumor suppressor genes, leading to the development, proliferation, and progression of leukemia cells [105]. In addition to miRNAs, small interfering RNAs (siRNAs) are another class of noncoding RNAs that play important roles in gene regulation. Both noncoding RNAs could be useful therapeutic agents for the treatment of cancer. These two classes of molecules share many similarities; both are short RNA molecules that silence genes at the posttranscriptional level by targeting messenger RNA (mRNA), yet their mechanisms of action and clinical applications are distinct. However, the major difference between siRNAs and miRNAs is that the former are highly specific, with only one mRNA target, whereas the latter have multiple targets. Therefore, the therapeutic approaches involving siRNAs and miRNAs greatly differ [106]. In this way, siRNAs could have utility in cancer therapy, but microRNAs are involved in the initiation, progression, and metastasis of ALL. Furthermore, miRNAs could be useful for discriminating between leukemia linages, immunophenotypes, molecular subtypes, high-risk-for-relapse groups, and poor/good responders to chemotherapy [105].

In this review, we identified several miRNAs involved in ALL relapse, and research on miRNAs may be helpful for developing future therapies for acute lymphoblastic leukemia (ALL). miR-124a and miR-155 have already been evaluated in relation to ALL, and results showed the possibility of therapeutic strategies for patients with ALL by the use of drugs that inhibit methylation and histone modifications or by the use of inhibitors specific to the target miRNAs [97,98]. However, guaranteeing the safety and effectiveness of these new therapies is necessary. Additionally, combining these therapies with conventional chemotherapy could facilitate treatment development, increase patient survival rates and prevent or cure refractory or relapsed ALL [107,108].

In conclusion, miRNAs are now considered promising indicators of prognosis in ALL, and studies based on patient data that were verified by in vitro experiments elucidated the role of miRNAs in molecular pathways other than relapse. Moreover, several of the molecular pathways underlying ALL relapse were clarified in this review.

## Figures and Tables

**Figure 1 ijms-25-00119-f001:**
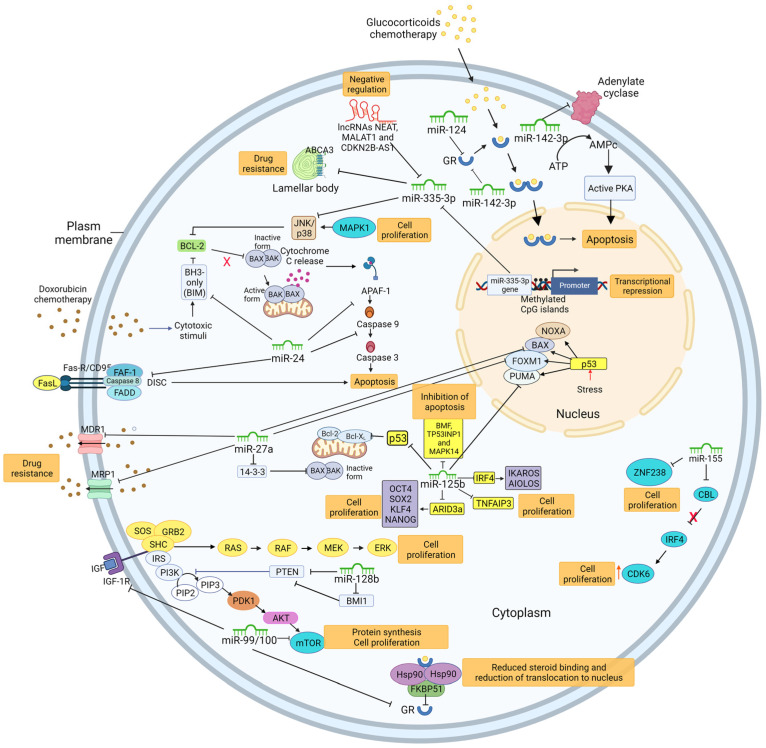
miRNAs involved in relapse in ALL patients and the related molecular mechanisms. miR-24; miR-27a; miR-99/100; miR-124; miR-1225b; miR-128b; miR-142-3p; miR-155; miR-335-3p. Inhibitory arrows show the targets of the miRNAs. The orange boxes show the processes associated with relapse. Red X indicates blockage of IRF4 degradation by CBL. The red arrow pointing up indicates increased CDK6 expression.

**Table 1 ijms-25-00119-t001:** Prognosis factors for acute lymphoblastic leukemia.

	Standard Risk Factors	High Risk Factors
Demographic and clinical features		
Age	1 year to <10 year	<1 year or ≥10 years
Sex	Female	Male
Clinical, biological or genetic features of ALL		
Blood count at diagnosis	Blood count < 50 × 10^9^ blood cells/mm^3^ per L for ALL-B and <100 × 10^9^ blood cells/mm^3^ per L for T-cell ALL	count ≥ 50 × 10^9^ blood cells/mm^3^ per L for B-ALL and ≥100 × 10^9^ blood cells/mm^3^ per L for T-ALL
Inmunophenotype	B-cell lineage	T-cell lineage
Cytogenetics features	Hyperploidy, ETS variant 6 (ETV6)- Runt-related transcription factor (RUNX), transcription factor 3 (TCF3)-PBX homeobox 1 (PBX1), trisomy of chromosomes 4, 10 or 17	Hypoploidy, Philadelphia chromosome positivity (breakpoint cluster region (BCR)-Abelson murine leukemia viral oncogene homolog 1 (ABL1)), MML TCF3-HLF rearrangement
Genomic features	Double homeobox 4 (DUX4) rearrangement (ERG deletion)	IKAROS family zinc finger 1 (IKZF1) deletions, Philadelphia chromosome type, myocyte enhancer factor 2D (MEF2D) rearrangements
Extramedullary disease	No	Yes
Response to treatment		
minimal residual disease (MRD)	Low minimal residual disease < 103 nucleated cells or undetectable	Persistence of minimal residual disease ≥ 103 nucleated cells

**Table 2 ijms-25-00119-t002:** Expression of miRNAs associated with relapse in ALL.

miRNA	Molecular Target	Drug resistance	Studies	Assay	Author
↓ miR-210	E2F3, RAD52	daunorubicine⁄dexametasone⁄ L-asparaginase/vincristine	Leukemia cells from patients with ALL Cell lines RS4;11 and REH	RT-qPCR	[93]
↑ miR-223	E2F1, E2A	Prednisolone	Leukemia cells from patients with ALL	Microarray	[14,15]
↓ miR-326	ABCA2	Prednisone/methotrexate/ asparaginase/6-MP	Leukemia cells from patients with ALL	RT-qPCR	[94]
↓ miR-652-3p	FOXK1	Vincristine/cytarabine	Leukemia cells from patients with ALL; cell lines Reh and RS4:11	Microarray	[3]
↑ miR-708	FOXO3	MTX/6-PM/daunorubicine/ prednisone/VCR/ L-asparaginase	Leukemia cells from patients with ALL and cell line Jurkat	Microarray	[14]
↑ miR-101–3p, ↑ miR-4774–5p, ↑ miR-1324, ↑ miR-631, ↑ miR-4699–5p, ↑ miR-922		Cisplatine/ carboplatine/methotrexate	Leukemia cells from patients with ALL	Microarray	[95]

↑ means overexpression of miRNA and ↓ means downregulation of miRNA.

## Data Availability

Not applicable.

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
