# Peer review of "The Role of miRNAs in Childhood Acute Lymphoblastic Leukemia Relapse and the Associated Molecular Mechanisms"

_ijms, 2023, doi:10.3390/ijms25010119_

Round 1

Reviewer 1 Report

Comments and Suggestions for Authors

The review article is very good and discussing an important and recent issue about miRNA but some comments are required:

1. in the introduction, it is better to mention the cost of the treatment and the socio-economic effect of ALL to intense the problems of the disease.

2. I suggest to clarify the risk-based approach of ALL therapy in a table form to make it easier and more understandable by the readers.

3. I also suggest to add a paragraph about the serious side effects of pediatric acute lymphoblastic leukemia therapy.

4. you have to check the superscription of numbers throughout the manuscript, for example 100x109 blood cells/mm3 per L. 9 must be superscript.

5. the resolution of Fig. 1 is not good enough and must be enhanced.

6. I suggest to add a heading about therapeutics that may target the discussed miRNAs. Numerous publications were elucidated the possible effect of several agents on miRNAs. Then, update the conclusion and future perspectives to include these therapeutics.

Comments on the Quality of English Language

The English language of the article is fine but minor check is required.

Author Response

December 15th, 2023

Prof. Dr. Alessandro Fatica Guest Editor,

Special Issue “The Emerging Role of RNA in Diseases and Cancers 2.0”. International Journal of Molecular Sciences,

MDPI,

Dear Professor Fatica,

Thank you for your letter dated 6 December 2023, concerning our manuscript, entitled “The role of miRNAs in childhood acute lymphoblastic leukemia relapse and the associated molecular mechanisms” by: Barrios-Palacios D et al., (Manuscript ID ijms-2771328).

After reviewing the content of your letter and the suggestions provided by the reviewers, we now submit a revised version of our manuscript, to which we have made several changes.

We include specific responses to the reviewer’s comments and the specific changes suggested by the reviewer, and attempt to clarify, as much as possible, each point raised. The new version of the manuscript incorporates the changes, which are shown in red in the text and with “track changes”. The manuscript has been revised and the English language improved and we sent the Editing certificate.

We really appreciate the suggestions provided by the reviewers, which greatly improved our manuscript.

We thank you in advance. Best regards,

Marco Antonio Leyva-Vázquez, PhD

Revisor 1

Comment 1. In the introduction, it is better to mention the cost of the treatment and the socio-economic effect of ALL to intense the problems of the disease.

Reply 1. We agree with that suggestion. In the Introduction we have mentioned the cost of treatment and the socioeconomic effect of ALL to intense the problems of the disease (Page 1, lines 42-52)

Comment 2. I suggest to clarify the risk-based approach of ALL therapy in a table form to make it easier and more understandable by the readers.

Reply 2. We have made the proposed change, since it improves the understanding of the text (Page 3, line 119)

Comment 3. I also suggest to add a paragraph about the serious side effects of pediatric acute lymphoblastic leukemia therapy.

Reply 3. We agree with that suggestion, and we´ve add an additional paragraph about the side effects of ALL therapy (Page 5, lines 176-199).

Comment 4. you have to check the superscription of numbers throughout the manuscript, for example 100x109 blood cells/mm3 per L. 9 must be superscript.

Reply 4. Thanks for the suggestion. We have reviewed the numbers in the article and corrected them. Now, 9 is in superscript (Table 1, on page 3)

Comment 5. the resolution of Fig. 1 is not good enough and must be enhanced.

Reply 5. We appreciate the comment. Figure 1 has been modified and now has a better resolution (Page 12). We will send Figure 1 in an attachment.

Comment 6. I suggest to add a heading about therapeutics that may target the discussed miRNAs. Numerous publications were elucidated the possible effect of several agents on miRNAs. Then, update the conclusion and future perspectives to include these therapeutics.

Reply 6. Thanks for the suggestion. A paragraph has been added to mention therapeutics that may target the discussed miRNAs (Page 10, lines 397-414), and we mention them in conclusion and futures perspectives. (Page 11, lines 466-474).

Comment 7. Comments on the Quality of English Language: The English language of the article is fine but minor check is required.

Reply 7. The manuscript has been revised and the English language improved and we sent the Editing certificate.

Reviewer 2 Report

Comments and Suggestions for Authors

The article entitles “The role of miRNAs in childhood acute lymphoblastic leukemia relapse and the associated molecular mechanisms” is the review specifically focused on the list of miRNAs in childhood acute lymphoblastic leukemia relapse and a future of way out. The article is well written and some crucial points might be be beneficial to improve it a bit

1.       Abstract is without any specific information. Authors should provide more information in the abstract section for the reader’s interest.

2.       Authors before jumping to the miRNA should provide the current treatment strategies for the relapse therapy of the disease.

3.       How the miRNA but not the siRNA or DNA is most significant in the lymphocytic leukemia?

4.       “ALL is a heterogeneous disease that comprises different immunophenotypic subtypes, and it is subclassified by identifying chromosomal and molecular abnormalities and gene expression patterns” – Authors need to provide in-depth details of the immunophenotypic subtypes, chromosomal and molecular abnormalities and gene expression patterns for the ALL.

5.       Authors need to provide a high-resolution Figure 1

6.       Is there any clinical trials with miRNA for ALL is under progress? Authors should provide ongoing clinical trials data for the miRNAs for the ALL (if any)

7.       Rewrite the future perspectives section and provide the advancement of the research in this filed. 

Comments on the Quality of English Language

A moderate grammatical editing is needed. 

Author Response

December 15th, 2023

Prof. Dr. Alessandro Fatica Guest Editor,

Special Issue “The Emerging Role of RNA in Diseases and Cancers 2.0”. International Journal of Molecular Sciences,

MDPI,

Dear Professor Fatica,

Thank you for your letter dated 6 December 2023, concerning our manuscript, entitled “The role of miRNAs in childhood acute lymphoblastic leukemia relapse and the associated molecular mechanisms” by: Barrios-Palacios D et al., (Manuscript ID ijms-2771328).

After reviewing the content of your letter and the suggestions provided by the reviewers, we now submit a revised version of our manuscript, to which we have made several changes.

We include specific responses to the reviewer’s comments and the specific changes suggested by the reviewer, and attempt to clarify, as much as possible, each point raised. The new version of the manuscript incorporates the changes, which are shown in red in the text and with “track changes”. The manuscript has been revised and the English language improved and we sent the Editing certificate.

We really appreciate the suggestions provided by the reviewers, which greatly improved our manuscript.

We thank you in advance. Best regards,

Marco Antonio Leyva-Vázquez, PhD

Revisor 2

 Comment 1. Abstract is without any specific information. Authors should provide more information in the abstract section for the reader’s interest.

Reply 1. We appreciate the suggestion. The abstract was modified to provide more information regarding the topic to be addressed in this review article (Page 1, line 17-30).

Comment 2. Authors before jumping to the miRNA should provide the current treatment strategies for the relapse therapy of the disease.

Reply 2. Thanks for the comment. We have added a title about the treatment strategies for the relapse therapy in ALL (Page 4, lines 158-175).

Comment 3. How the miRNA but not the siRNA or DNA is most significant in the lymphocytic leukemia?

Reply 3. We appreciate this suggestion and we´ve added a paragraph on the significant importance of miRNAs with respect to DNA and siRNAs (Page 11, lines 449-465)

Comment 4. “ALL is a heterogeneous disease that comprises different immunophenotypic subtypes, and it is subclassified by identifying chromosomal and molecular abnormalities and gene expression patterns” – Authors need to provide in-depth details of the immunophenotypic subtypes, chromosomal and molecular abnormalities and gene expression patterns for the ALL.

Reply 4. We agree with that suggestion. So, we provided more details of the immunophenotypic subtypes, chromosomal and molecular abnormalities and gene expression patterns for the ALL (Page 2, lines 87-107)

Comment 5. Authors need to provide a high-resolution Figure 1

Reply 5. We appreciate the comment. Figure 1 has been modified and now has a better resolution (Page 12). We will send Figure 1 in an attachment.

Comment 6. Is there any clinical trials with miRNA for ALL is under progress? Authors should provide ongoing clinical trials data for the miRNAs for the ALL (if any)

Reply 6. Yes, there are clinical trials in ALL. However, they are not in progress. Data from clinical trials have been provided in this review (Page 10, lines 415-426)

Comment 7. Rewrite the future perspectives section and provide the advancement of the research in this filed.

Reply 7. We appreciate the comment. The conclusion and future perspectives section has been modified, and is now more enriched by the readers (Page 10, lines 428-478).

Comment 8. Comments on the Quality of English Language: A moderate grammatical editing is needed.

Reply 8. The manuscript has been revised and the English language improved and we sent the Editing certificate.

Round 2

Reviewer 1 Report

Comments and Suggestions for Authors

Thank you for your responses.

Reviewer 2 Report

Comments and Suggestions for Authors

Dear Authors 

The article is well revised. 

Comments on the Quality of English Language

No need